

# LiBiNorm: an htseq-count analogue with improved normalisation of Smart-seq2 data and library preparation diagnostics

Nigel P. Dyer[1], Vahid Shahrezaei[2] and Daniel Hebenstreit[1]

[1] School of Life Sciences, University of Warwick, Coventry, UK
[2] Department of Mathematics, Imperial College London, London, UK

## ABSTRACT

Protocols for preparing RNA sequencing (RNA-seq) libraries, most prominently "Smart-seq" variations, introduce global biases that can have a significant impact on the quantification of gene expression levels. This global bias can lead to drastic over- or under-representation of RNA in non-linear length-dependent fashion due to enzymatic reactions during cDNA production. It is currently not corrected by any RNA-seq software, which mostly focus on local bias in coverage along RNAs. This paper describes LiBiNorm, a simple command line program that mimics the popular htseq-count software and allows diagnostics, quantification, and global bias removal. LiBiNorm outputs gene expression data that has been normalized to correct for global bias introduced by the Smart-seq2 protocol. In addition, it produces data and several plots that allow insights into the experimental history underlying library preparation. The LiBiNorm package includes an R script that allows visualization of the main results. LiBiNorm is the first software application to correct for the global bias that is introduced by the Smart-seq2 protocol. It is freely downloadable at http://www2.warwick.ac.uk/fac/sci/lifesci/research/libinorm.

## INTRODUCTION

RNA sequencing (RNA-seq) is now widely used for determining gene expression levels. While originally developed for assessing average gene expression over a large number of cells from a cell culture or tissue sample, it is increasingly being used for measuring gene expression within individual cells (*Tang, Lao & Surani, 2011*).

Most sample preparation protocols for RNA-seq involve RNA isolation and reverse-transcription, followed by fragmentation, amplification, and sequencing, not necessarily in this order; this is followed by aligning the sequencing reads to a reference sequence and allowing the reads to be associated with a gene or transcript so that expression levels can be calculated.

A major issue with most RNA-seq strategies is the question of how to translate sequencing read numbers into abundances of the original RNAs.

There are currently two main strategies for using RNA-seq data to quantify transcript expression levels: (i) tagging RNAs with unique molecular identifiers (UMIs), which allows

Corresponding authors
Nigel P. Dyer,
nigel.dyer@warwick.ac.uk
Daniel Hebenstreit,
d.hebenstreit@warwick.ac.uk

labelling and in turn counting absolute numbers of original RNA molecules (*Islam et al., 2014*); (ii) sequencing fragments derived from the whole RNA length, which prevents UMI usage, since there is no way to identify fragments stemming from the same original RNA. However, a larger amount of the original RNAs is thus converted into useful information, increasing sensitivity compared to UMI-based approaches (*Ziegenhain et al., 2017*).

The first strategy is implemented by protocols such as CEL-seq (*Hashimshony et al., 2012*) and is arguably more precise by design, yet usually includes more experimental steps and loses all information on the 5′ RNA sequence. The second strategy is implemented by protocols such as RNA-fragmentation (e.g., Illumina TruSeq), Smart-seq (*Ramskold et al., 2012*), Quartz-seq (*Sasagawa et al., 2013*), random priming (*CSHL, 2005*), and Smart-seq2 (*Picelli et al., 2014*). Both strategies are popular for RNA-seq in general, and will probably remain so in light of their complementary nature. Smart-seq2 in particular is increasingly employed for single cell RNA-seq.

The default assumption is that there is a linear dependence between the expected read numbers and the RNA length, as embodied by length-normalizing measures such as Fragments per Kilobase Million (FPKM) (*Trapnell et al., 2010*) and Transcripts per Kilobase Million (TPM) (*Li et al., 2010*). Deviations of the data from this notion are common, mostly due to the complexity of the experimental protocols, and software has been developed to correct for such "bias."

However, virtually all bias correction efforts target deviations from a uniform read distribution along RNAs, also referred to as "coverage." It was only recently recognized that cDNA conversion of protocols such as Smart-seq2 not only affects coverage, but introduces a global, length-dependent bias that is potentially orders of magnitude stronger than "local" coverage bias. We established this as a problem in our previous work (*Archer et al., 2016*), which is confirmed by other studies (*Phipson, Zappia & Oshlack, 2017*).

While local biases are mostly related to DNA sequence and affect the likelihood that the DNA will fragment or the likelihood that primers will associate at a particular location, we showed, based on a set of abstracted models, that global biases arise from aspects of polymerase processivity during cDNA conversion; the latter are less dependent on the DNA sequence and often cause a variation in read coverage along the transcript, but not always. This resulting length-dependent overall bias for a transcript can introduce significant errors when calculating relative expression levels of genes with different transcript lengths. In fact, the read numbers obtained for longer RNAs in Smart-seq protocols deviate so much from FPKM/TPM measures that the assumption of linear scaling appears wrong altogether and will lead to an underestimation of long genes' expression levels (*Archer et al., 2016*).

Several methods and software releases have been published which attempt to cater for different sources of bias that can occur when assessing expression levels using RNA-seq data. To our knowledge, none of these consider the global bias that we address here. Instead, they mostly assume an overall linear scaling between read numbers

and RNA lengths and focus on local deviations from uniform coverage. These are documented in more detail in the Supplemental Information Section 1.

Here, we present the standalone tool "LiBiNorm" that builds on our previous work and permits easy and user-friendly implementation of analysis strategies aimed at global bias. Its main target is Smart-seq2 data, but it can be applied broadly to diagnose library preparation characteristics underlying different types of datasets.

## MATERIALS AND METHODS

### Implementation

We have developed LiBiNorm (library bias normalization), a software package that can quantify the degree of global bias present in a Smart-seq2 RNA-seq dataset using the framework described in *Archer et al. (2016)* and then produce gene expression data that has been normalized to compensate for this bias. In contrast to Archer et al. which presents results that were derived based on custom scripts in several different programing languages, LiBiNorm combines the analyses strategies used in that work into a single tool. LiBiNorm is a command line program available for Windows, Linux and Apple Mac, together with an optional R script for producing graphical output.

LiBiNorm takes aligned bam-format Smart-seq2 RNA-seq data as input together with a reference genome annotation in gtf or gff3 format. The RNA-seq data can be single or paired end and must be aligned to a reference genome by an aligner such as Tophat2 (*Kim et al., 2013*), its successor HISAT2 (*Pertea et al., 2016*) or STAR (*Dobin et al., 2013*) that will align reads across intron splice sites. Wherever possible, LiBiNorm's command line parameters and algorithm for associating reads with genes mirror those of htseq-count (*Anders, Pyl & Huber, 2015*). However, unlike htseq-count, LiBiNorm does not require large RNA-seq datasets to be ordered by genome position prior to being analyzed.

### Models

LiBiNorm associates individual reads with the transcripts associated with each gene and from their distribution determines parameters that are appropriate for normalizing the effect of bias on gene expression values. These parameters are based on mathematical models, as introduced in (*Archer et al., 2016*), which capture characteristics of different library preparation protocols. The models are explained in more detail in Supplemental Information Section 2. In brief, six alternative abstracted models are used (Models A, B, C, D, BD, E), the first four of which are only included for completeness and permit modeling of various aspects/steps of different protocols (*Archer et al., 2016*). Model BD is appropriate for the Smart-seq2 protocol and is thus used by default. Model E corresponds to random-priming based library preparation and can be used to study and correct biases of these, albeit this type of protocol is rarely used now. The models predict mathematically the expected protocol-specific read distributions for different RNA lengths, and are tested by LiBiNorm regarding their agreement with the supplied RNA-seq data.

Using the gene transcript lengths that can be derived from the genome annotation, it determines parameters corresponding to the model using non-overlapping, single

isoform genes only and then produces a modified TPM measure of the gene expression which has been normalized to compensate for the global bias.

It is possible to specify that an alternative to the default BD model be used for bias normalization, or that all models should be evaluated and their goodness-of-fits based on log likelihoods used to select the best model for a specific data set.

LiBiNorm is in principle suited to all Smart-seq data. However, we have found that local bias in Smart-seq1 data is so strong that it impedes global bias correction or masks its effects. This is probably due to RNA secondary structures that cause erroneous reverse transcription initiation and termination, which disturbs the read distribution, thus frustrating accurate determination of the model parameters. This problem is overcome in Smart-seq2 through the addition of betaine (*Picelli et al., 2014*).

## Input data

The primary input data for LiBiNorm is a bam file containing aligned RNA-seq data. Core command line options allow identifying the type of RNA-seq data (stranded or non-stranded, ordered by name or genomic position), choosing the genomic feature to which the reads should be aligned, and setting the rules governing whether a read is associated with a feature or not. As well as the bam file, LiBiNorm also requires a gene annotation file that allows the reads to be assigned to specific genes which can either be in gff3 or gtf format.

The LiBiNorm takes as input single or paired end RNA-seq data in bam format after having been aligned to a complete reference genome. LiBiNorm must be informed if the data is paired end and not ordered by read name. RNA-seq protocols can generate reads that are aligned with, or are in the opposite sense to, the sense of the original RNA, or alternatively may not indicate the original sense at all. The read sense status must be provided to LiBiNorm through the associated command line option.

## Assignment of reads to genes

The bias correction performed by LiBiNorm is a function of the lengths of the transcripts and the distribution of the RNA fragments within these transcripts.

By default, LiBiNorm associates reads with the combined exons of all of the transcript variants of a gene. The length of the mRNA transcript is assumed to be the total length of the combined exons, and the position of a read is taken as the position within this total length. The presence of some transcript variants with retained introns is recorded in some reference annotations, but such variants are unusual and give misleading gene lengths and read positions, so are discarded by LiBiNorm.

LiBiNorm adopts the same three alternative algorithms for determining whether a read is associated with a feature as is available in htseq-count. However, the LiBiNorm default is the "strict" option, which requires the read to lie completely within annotated exons and not extend outside the exons in order to be associated with a gene. This avoids incorrect association of reads with small RNA genes which can provide misleading information when estimating model parameters.
## Parameter estimation

Our models are specified by up to five parameters that need to be estimated for each given dataset (*Archer et al., 2016*). Parameter estimation is performed only using reads associated with the subset of genes or transcripts that are unique, that is, do not overlap other genes or transcripts. This avoids systematic errors that may arise during the process of assigning reads to genes where ambiguous reads are discarded. Up to 200 reads are randomly selected for each of these genes and these are used in a two-stage process for determining model parameters. The initial stage is to use the Nelder-Mead simplex method (*Nelder & Mead, 1965*) to identify a suitable starting point for a set of Metropolis-Hastings Monte Carlo Markov chain (MCMC) (*Wilkinson, 2011*) runs.

LiBiNorm ignores mRNAs longer than 20 kb in this process in order to limit the potential influence of very long transcripts with spurious reads. We have found the default cutoff of 20 kb to be adequate for all our analyses, however a command line option permits changing this setting. We have also included default settings for the maximum total reads ($10^9$) and reads from each gene (200) that are selected and the number and lengths of MCMC runs (10 and 200, respectively) that are performed. These choices of settings combine high precision and quick run time, but can be changed by the user. Finally, the initial Nelder-Mead step is performed by default as it reduces the fraction of MCMC runs that get trapped in local modes; it can be switched off by command line option, though, if random starting points are preferred.

Importantly, the parameters $d$, $h$, $t_1$, $t_2$, and $a$ that are associated with the models (all five with BD; the first four with E) are inferred from the data and make it possible to "reverse-engineer" aspects of the sample preparation process. These parameters correspond to different aspects of the sample preparation process. This includes reduced fragmentation efficiencies (by factor $d + 1$) at fragment ends (over distance $h$ from ends), reverse transcriptase and second-strand polymerase processivities ($1/t_1$ and $1/t_2$, respectively), and PCR efficiency of Smart-seq protocols (*Archer et al., 2016*).

It is also possible to specify that one of the other models, or all six models should be used for comparison purposes.

## Output

Bias corrected gene expression data is sent to the output stream per default, allowing it to be piped to another process or a file. Alternatively, a filename can be specified, which produces a file in tab separated variable format.

LiBiNorm can also produce simple read count data from the RNA-seq data that can be used by packages such as DESeq2 (*Love, Huber & Anders, 2014*) for calculating differential expression. This avoids the need to use both LiBiNorm and another software package such as htseq-count in the analysis pipeline. LiBiNorm has other advantages over htseq-count in that it is faster than htseq-count by virtue of being a monolithic compiled executable rather than a hybrid of compiled and python interpreted code; on a 64 bit linux server, the SRR1743160 sample used in this paper (see below) as a representative dataset took 1,613 s with htseq-count, whereas LiBiNorm in htseq-count compatible mode took 125 s. The output files are identical.

**Table 1 Output files.**

| Filename | Description |
|---|---|
| <countfilename>*<br>  <fileroot>_expression.txt** | Main output file with raw counts, gene length, global bias, and bias-corrected, normalized TPM expression levels for all genes. |
| <fileroot>_bias.txt** | Consolidated data indicating the distribution of reads within the transcripts. The transcripts are ordered by length and then grouped into 500 roughly equal bins. The file gives the average gene length and a histogram of the read distribution for each of the bins. |
| <fileroot>_norm.txt** | Parameter estimates and bias predicted by the model as a function of selected transcript lengths, which forms the basis of the normalization which is applied by the model. |
| <fileroot>_results.txt** | Provides detailed information relating to the parameter estimation process including the results from each of the MCMC runs used to generate these results and an indication of the spread of the parameter estimates that were obtained from these runs. |
| <fileroot>_distribution.txt** | Histogram of the read distribution within the transcripts for five different groups of transcripts each centered on a specific transcript length, together with the distribution predicted by the model for these lengths. |

**Notes:**
* created as a result of the —c command line option. Otherwise these data are sent to the program standard output.
** only created if the —u command line option is used.

The commands used were:

```
LiBiNorm count -z -i gene_id -s no -c <countfilename> <fileroot>.bam
```
"../Drosophila_melanogaster.BDGP6.91.gtf"

and

```
htseq-count -i gene_id -f bam -s no <fileroot>.bam
```
"../Drosophila_melanogaster.BDGP6.91.gtf" > <countfilename>

Library bias normalization can, if specified through the command line, generate additional files that contain more information about the bias analysis it has performed. The full set of output files is shown in Table 1.

The output files can be processed by an R script that forms part of the LiBiNorm software package in order to produce various graphical representations of the bias and parameter estimation process. The plots include a heatmap that visualizes the global bias and is by itself a highly useful tool for RNA-seq QC and diagnostics.

## Software architecture

LiBiNorm is implemented in C++. It uses a slightly modified version of the bamtools library for reading bam files and uses the standard C++11 template library. The associated makefile supports compilation using gcc or clang in order to generate executables for Linux or macOS. A Microsoft Visual Studio solution file supports compilation for Windows platforms, for which the source for the zlib compression library is also required.

# RESULTS

We illustrate usage of LiBiNorm based on datasets from a *Drosophila* study that compares different RNA-seq protocols. This study is particularly well suited for our analyses, as it includes (virtually) global bias free RNA-fragmentation datasets (TruSeq) and Smart-seq2 data *Combs & Eisen (2015)*. We therefore downloaded datasets SRR1743153–SRR1743166 from the short read archive (SRA; https://www.ncbi.nlm.nih.gov/sra). Comparing these datasets will be used to demonstrate the reduction in global bias provided by LiBiNorm compared to that of other software packages.

## Processing

The reads from the 14 Smart-seq2 datasets SRR1743153–SRR1743166 were aligned using HISAT2 (*Kim, Langmead & Salzberg, 2015*) version 2.0.5 to the bdgp6_tran reference genome downloaded from the HISAT2 website (http://ccb.jhu.edu/software/hisat2/index.shtml). The output was converted to bam file format using samtools version 1.3.1 (*Li et al., 2009*) as follows:

```
hisat2 p 10 -x bdgp6_tran/genome_tran -U <fileroot>.fastq -S
<fileroot>.sam
    samtools view -b <fileroot>.sam > <fileroot>.bam
```

LiBiNorm count was then used to produce bias corrected expression data against the Drosophila_melanogaster.BDGP6.91 reference genome obtained from http://ftp.ensembl.org/pub/release-91/gtf/drosophila_melanogaster/:

```
LiBiNorm count -p 3 -i gene_id -u <fileroot> -s no -c <countfilename>
<fileroot>.bam Drosophila_melanogaster.BDGP6.91.gtf
```

This creates the full set of output files that contain the normalized expression values and details of the bias in the dataset and the process of parameter estimation.

Sample SRR1743160 (Smart-seq2) was analyzed using the LiBiNorm "–*f*" option which produces results for the full set of six models that are described in *Archer et al. (2016)*:

```
LiBiNorm count -p 3 -f -i gene_id -u <fileroot> -s no -c <countfilename>
<fileroot>.bam Drosophila_melanogaster.BDGP6.91.gtf
```

## Visualisation

LiBiNormPlot.R was then used to produce plots from the data files and which are shown in Fig. 1.

```
Rscript LiBiNormPlot.R SRR1743160
```

Plots in Figs. 1A and 1E allow displaying and assessing the global bias in the SRR1743160 data in two different formats. The heatmap (Fig. 1A) displays (normalized) read density along transcripts by color intensity, with transcripts aligned at 5′ and 3′ ends and ordered from shortest (top) to longest (bottom); an even coverage along transcripts would have a uniform orange color, whereas an irregular clustering of reads would feature as black spots interspersed with white low-density regions. Similarly, (Fig. 1E) shows

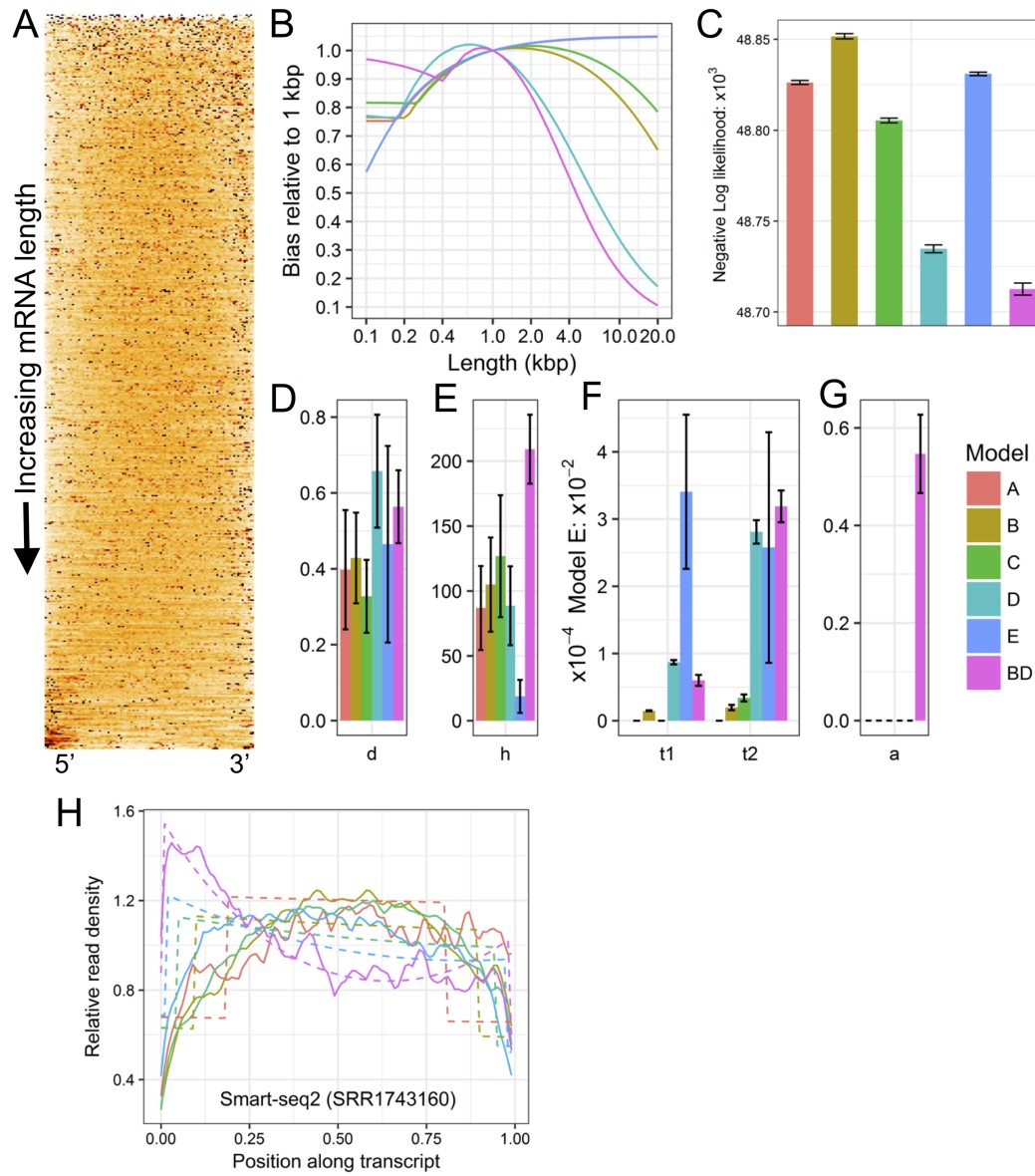

**Figure 1 Example plots of read bias (SRA accession SRR1743160) produced with LiBiNorm.** (A) detected transcripts are aligned at 5′ and 3′ ends and ordered by length, shortest on top. Read density along RNAs is indicated by color intensity (the darker, the higher). (B) predicted bias for each model as a function of transcript length: bias relative to a linear length model. (C) comparison of negative log likelihood values (the lower the better the fit) for each of the six models with parameters determined for the SRR1743160 dataset. (D–G) estimated model parameter values $d$, $h$, $t_1$ & $t_2$, and $a$, respectively. See text for interpretation of parameters. (H) read coverages along transcripts aligned at 5′ and 3′ ends and separated into different length classes (colors). The experimental data and model fits are shown separately as solid and dashed lines (fit of model BD), respectively.

"sections" through the heatmap, that is, coverage as lines for transcripts grouped into different approximate length classes, with dashed lines indicating the fitted model's (BD in this case) predictions. Both types of plots clearly visualize how read coverage varies with gene length in a non-linear fashion typical for Smart-seq protocols. Figures 1D–1G show the bias predicted for all six models, the relative likelihood of each of the models and

**Table 2 LiBiNorm output for SRR1743160 and model BD parameter and bias estimates.**

| Model | Goodness-of-fit (log likelihood) | Parameter estimates | | | | |
|---|---|---|---|---|---|---|
| | | $\log_{10}(d-1)$ (ratio) | $\log_{10}(h)$ (bases) | $\log_{10}(t_1)$ (bases$^{-1}$) | $\log_{10}(t_2)$ (bases$^{-1}$) | $a$ (fraction) |
| BD | 48722 | −0.103 | 1.89 | −4.27 | −3.52 | 0.59 |

| Length (bases) | 100 | 120 | 140 | 160 | 180 | 200 | 220 | ... | 19,800 | 19,900 | 20,000 |
|---|---|---|---|---|---|---|---|---|---|---|---|
| Bias (ratio) | 0.763 | 0.759 | 0.755 | 0.772 | 0.830 | 0.875 | 0.911 | ... | 0.097 | 0.097 | 0.096 |

the parameter estimates (see next section). The (negative −) log likelihoods suggest that Model BD is the most appropriate for the Smart-seq2 data as it is lowest Fig. 1C.

## Parameter interpretation

<fileroot>_norm.txt contains parameter estimates and predicted bias relative to a linear model for each of the six models. The Model BD $\log_{10}$ (except $a$) parameter estimates and predicted bias for SRR1743160 is shown in Table 2.

These parameters allow insights into characteristics of the library preparation process and can be interpreted in the following way: the library's fragmentation/tagmentation efficiency was reduced by $10^{-0.103} + 1 = 1.788$ ($d$), over a length of $10^{1.89} = 78$ bases ($h$) at each end of the fragments. The first- and second-strand processivities (inverses of $t_1$ and $t_2$) are $10^{4.27} = 18.6$ kb ($t_1$) and $10^{3.52} = 3.31$ kb ($t_2$), respectively. Finally, PCR selection has increased the proportion of complete second strands to 59% ($a$). While these values appear sound, this diagnostics information potentially allows troubleshooting; for example, processivities or PCR efficiency could be increased by optimizing reaction conditions etc.

The bias vs. length table (Table 2) further indicates the error produced when using conventional measures of transcription that do not account for global bias. If TPM or FPKM were used to compare the expression levels of 200 and 20,000 bp long transcripts, they would be in error by a factor of 0.875/0.096 or 9.11.

## Evaluation of bias removal

TruSeq data is virtually bias-free, since it is based on RNA fragmentation, which drastically reduces the effects of the cDNA conversion. We used the four TruSeq samples in the Combs et al. data as the gold standard for mRNA expression levels; other popular benchmarking methods, such as qPCR quantification are inadequate, as they are subject to the same global cDNA bias.

The approach adopted was to determine the correlation of the $\log_2$ expression levels, expressed as TPM (*Li et al., 2010*) between Smart-seq2 and TruSeq data. If the global bias in Smart-seq2 data is reduced successfully, the correlation should improve.

The <fileroot>_counts.txt file was used to determine the correlation between the per-gene expression levels calculated using the TruSeq RNA-seq data and the Smart-seq2 data. For the latter, we used either the standard TPM values or the bias-corrected TPM.

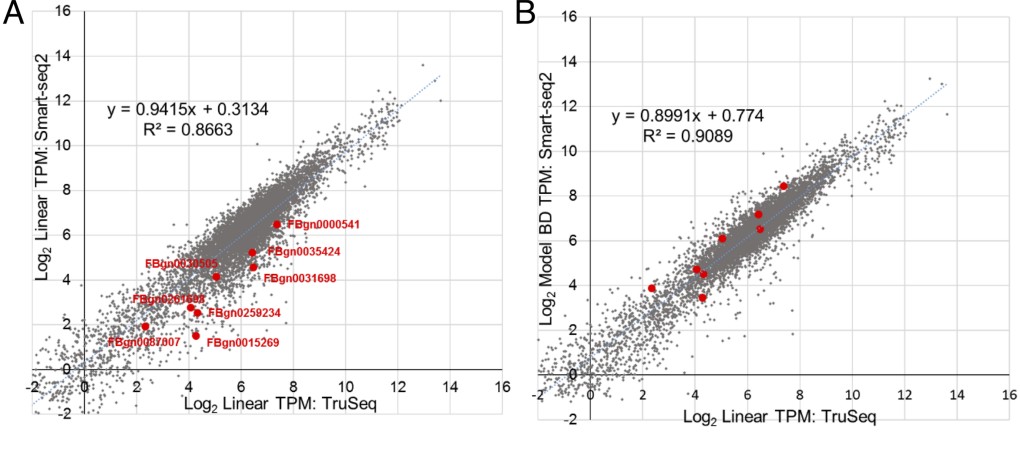

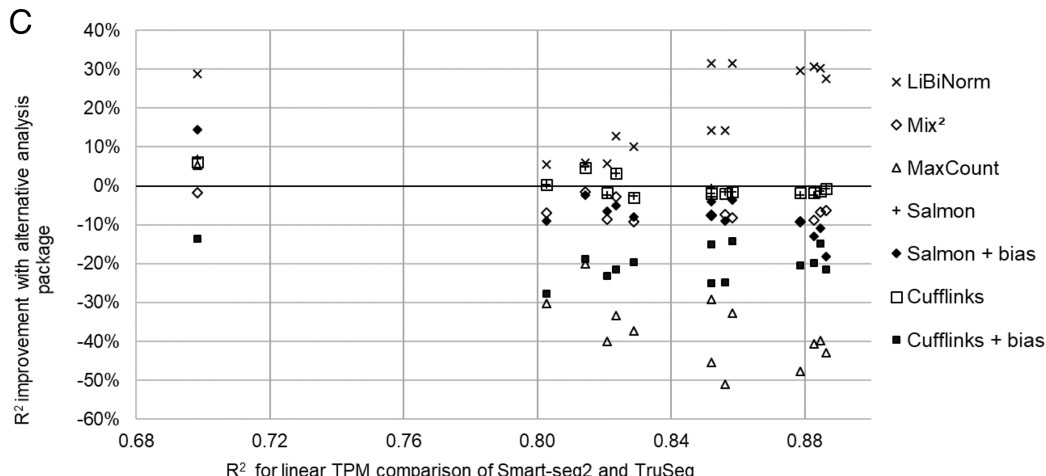

**Figure 2 Evaluation of bias correction.** (A) scatter plot of gene expression values derived from RNA-seq using TruSeq (SRR1743167) and Smart-seq2 (SRR1743160) based on conventional (linear; equivalent to FPKM) TPM. (B) same as (A), but using LiBiNorm (Model BD) to calculate TPM for the Smart-seq2 sample, which improves the $R^2$ compared to conventional TPM. Red dots mark genes with mRNA lengths between 10 and 10.1 kb in length, showing how the bias correction compensates for the underestimated expression levels of these genes. (C) change of $R^2$ (%; $y$-axis) when systematically comparing gene expression for Smart-seq2 and TruSeq protocols compared to a linear TPM reference ($x$-axis). An average across the four TruSeq samples is plotted for each of the 14 Smart-seq2 samples for each of the software packages as indicated.

Using the same Smart-seq2 sample as before (SRR1743160) and its fitted model (BD), we demonstrate based on scatter plots how correlation improves after bias correction (Figs. 2A and 2B). A selection of long genes (10.05 ± 0.05 kb; red dots) illustrates underestimation of their expression levels with Smart-seq2 and conventional, linear quantification models.

To test this systematically, we calculated the $R^2$ of this correlation for all combinations of the 14 Smart-seq2 and four TruSeq samples, in each case only comparing those genes with more than 10 reads in both samples. We saw an average improvement of over 21% across all of the Smart-seq2 samples (Fig. 2C, "×" symbols).

For comparison, we performed the same process using four of the software packages which contain some degree of bias removal: Cufflinks (*Roberts et al., 2011*), Salmon (*Patro et al., 2017*), MaxCount (*Finotello et al., 2014*) and Mix$^2$ (*Tuerk, Wiktorin & Güler, 2017*). Both Salmon and Cufflinks contain bias removal options and these packages were evaluated with and without bias removal (Fig. 2C) (See Supplemental Information Sections 3 and 4 for detailed methods and results). In virtually all cases, the software packages worsened correlations between the datasets; Salmon and Cufflinks gave almost identical results as each other and their average correlation was the same as when linear TPM is used. Correlation deteriorated when the bias normalization options were enabled for both Salmon (−6.5%) and Cufflinks (−20%). Mix$^2$ showed an average 6.6% poorer correlation than the linear TPM reference, and maxCount was 34.7% poorer (Fig. 2C). LiBiNorm's improvement of the $R^2s$ over the alternative tools was statistically significant in all cases ($P < 1.1 \times 10^{-5}$; one-sided paired *t*-test).

## CONCLUSIONS

LiBiNorm is a multi-platform software application designed to identify, visualize, and correct global biases in RNA-seq data that are introduced by the library preparation protocols.

LiBiNorm functions by learning parameters from the datasets that allow reverse engineering parts of the library preparation history. The parameter estimates are characteristic for the library preparation protocol and allow LiBiNorm to infer the protocol that was used. This permits insights and diagnostics of the sample preparation history and enables LiBiNorm to calculate and apply the appropriate bias correction, which is most relevant for the Smart-seq2 protocol. A benchmarking effort confirms that LiBiNorm is superior to other available software for this purpose.

A companion R script is able to produce graphical representations of the results and analyses LiBiNorm generates. LiBiNorm was designed to be consistent with the gene expression calculations provided by htseq-count (*Anders, Pyl & Huber, 2015*) and includes an htseq-count compatible mode.

### Funding

This work has been supported by BBSRC research grants BB/L006340/1 and BB/M017982/1. The funders had no role in study design, data collection and analysis, decision to publish, or preparation of the manuscript.

### Grant Disclosure

The following grant information was disclosed by the authors:
BBSRC research: BB/L006340/1 and BB/M017982/1.

### Competing Interests

The authors declare that they have no competing interests.

## Author Contributions

- Nigel P. Dyer analyzed the data, prepared figures and/or tables, authored or reviewed drafts of the paper, approved the final draft.
- Vahid Shahrezaei analyzed the data, authored or reviewed drafts of the paper.
- Daniel Hebenstreit analyzed the data, prepared figures and/or tables, authored or reviewed drafts of the paper, approved the final draft.

## Data Availability

GitHub: https://github.com/NigelDyer/LiBiNorm.

## Supplemental Information

Supplemental information for this article can be found online at http://dx.doi.org/10.7717/peerj.6222#supplemental-information.

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
