# Peer review of "LiBiNorm: an htseq-count analogue with improved normalisation of Smart-seq2 data and library preparation diagnostics"

_PeerJ, doi:10.7717/peerj.6222_

## Round 0.1 · original submission · Minor Revisions

Please make sure you address all reviewers comments, especially on the validity of the findings. Please provide more information in comparison with other approaches.

·

Basic reporting

no comment

Experimental design

no comment

Validity of the findings

no comment

Additional comments

This paper described software implementing bias correction methods proposed in the authors' earlier paper [1]. The method described in that paper (and summarized in the supplemental material) here is compelling and raises interesting questions, but the scope of this paper is to describe the tool LiBiNorm they devolved to make that methodology usable by investigators, so I'll similarly restrict the scope of the review to the tool without digging into the underlying model and findings in [1].

The paper does a good job of describing the background and how the tool works. It additionally shows a benchmark comparing bias corrected Smart-seq2 to TruSeq. There are different ways that estimates of relative expression can be compared, but the approach here isn't unreasonable. The results are somewhat surprising though, as they show bias correction with other methods often leading to diminished accuracy. However, I haven't seen this comparison made before (that is, Smart-seq2 vs TruSeq), so it doesn't necessarily contradict previous papers that show improvements in accuracy between bulk RNA-Seq and qPCR with bias correction.

I downloaded and compiled the software without issue. It's well documented on the website and includes a open source license (and the licenses of included libraries). To test the program I ran it on some BAM files from [2]. The program was quite efficient, e.g. processing 27 million alignments in less than a minute. I didn't attempt to independently benchmark the method, but did inspect the output files, which looked reasonable.

I have some minor concerns about the implementation. The models are fit "using non-overlapping, single isoform genes only". I ran the method on human data, using Ensembl gene annotations, which annotates multiple isoforms for most protein coding genes. It leaves me a little nervous that these are excluded, as the remaining genes may not be characteristic the sample as a whole (e.g. if it's mostly non-coding, or lincRNA). Still, this may not matter if, as the authors posit, the global bias is a function of length primarily.

This is a minor point of concern that I hope the authors will consider, but shouldn't hold up the manuscript. Overall, it's a well written paper, and well constructed tool, and I recommend accepting it for publication.

[1] Archer, Nathan, Mark D. Walsh, Vahid Shahrezaei, and Daniel Hebenstreit. 2016. “Modeling Enzyme Processivity Reveals That RNA-Seq Libraries Are Biased in Characteristic and Correctable Ways.” Cell Systems 3 (5): 467–79.e12.

[2] Tasic, Bosiljka, Vilas Menon, Thuc Nghi Nguyen, Tae Kyung Kim, Tim Jarsky, Zizhen Yao, Boaz Levi, et al. 2016. “Adult Mouse Cortical Cell Taxonomy Revealed by Single Cell Transcriptomics.” Nature Neuroscience 19 (January): 335.

Reviewer 2 ·

Basic reporting

The authors present a well written and clear description of their software. However, the title and text are somewhat misleading when it comes to the strong focus on comparing LiBiNorm to HTSeq, since in fact LiBiNorm seems to be more akin to a specific tool implemented using the HTSeq library, i.e. the htseq-count script, and not a software library intended as a tool for building bioinformatics applications.

I have seen that the authors do make the correct comparison on their website and would encourage them to amend the article accordingly. It would seem that the authors dwell on this comparison maybe more than necessary, since LiBiNorm is first and foremost a modelling Software implementing a specific bias model and I would have expected that the authors would put a comparison of the successful bias modelling to the performance of other bias-removal strategies as shown in e.g. Figure 2c more into the foreground.

A few sections could benefit from clarification, e.g. on line 156 the authors state that LiBiNorm operates in strict mode by default, requiring each read to be fully contained within an annotated exon, this seems somewhat contradictory to the statement on lines 99-101 where a splice-aware aligner is required, which would imply that reads can span exons and would be considered in that case.

On Lines 184-186 the authors generally state that LiBiNorm is faster than HTSeq (I presume they talk about htseq-count) because it is compiled. Has this gain in performance been quantified? If so, it should be reported here, otherwise this part seems unnecessary since it is not clear if there is any tangible time benefit for people that build time-critical sequencing pipelines. A sufficiently more complex or not particularly well optimised compiled program might even be slower than the htseq-count script, or any other interpreted script for that matter.

The software is presented well on a dedicated webpage. I would suggest to use higher quality images in the section about Graphing results (https://warwick.ac.uk/fac/sci/lifesci/research/libinorm/graphs/), currently the plots have a very poor resolution. It might be worthwhile to consider moving/duplicating this information in a project-associated GitHub wiki to ensure it will be available even if the software is not actively developed anymore / the authors move on in their academic careers etc.

Experimental design

The methods are generally well described with some clarifications needed as specified in section 1 (Basic reporting).

Validity of the findings

As mentioned above the strong focus on comparing to HTSeq (or rather htseq-count) seems to take away focus from the comparison of the bias removal to other approaches, I would suggest to focus more on this aspect (although that might be already covered to a relatively large degree in the previous publication that describes the models themselves).

·

Basic reporting

- English writing is clear and acceptable
- Structure of this paper needs to be improved. Example: some short paragraphs could be combined to a long paragraph with much information.
- Full set of output files (page 8) should be displayed as Table or the authors should use another way to display. Because it is not scientific now.
- Tables in the manuscript aren't displayed in a scientific way.
- It should be better if the authors provide the references for some tools they used, i.e., Tophat2, STAR, or Nelder-Mead …

Experimental design

How this work different with previous work (Archer et al. 2016), authors need to discuss more this issue?

Validity of the findings

- Can the authors test their method on a general dataset? Because now the methods only performed well on a specific dataset
- Where is the primary input data authors retrieved?
- How to choose the optimal parameters in this work?
- Why the authors used SRR1743160 data? Any specific reason?
- The authors should compare with other works more detail. For example, authors can use a paired t-test to test the significances among the works.

Additional comments

The authors should discuss more the differences between the current work and their previous work. They also need to discuss the detailed comparison with other works. It will help their work more strong and enough to meet the requirement of publication.

---

## Round 0.2 · accepted · Accept

Please note that PeerJ does not provide editorial services so please make sure that all information in your manuscript is accurate while entering the production process.

# Reviewer 2 ·

Basic reporting

The manuscript is now more clear and I have no further comments.

Experimental design

I have no comments in this section.

Validity of the findings

I have no further comments.

·

Basic reporting

No comment

Experimental design

No comment

Validity of the findings

No comment

Additional comments

The authors have already addressed all of my comments, therefore, I think the manuscript now meets the requirement of PeerJ for publication.